# Inhibition of JAK1,2 Prevents Fibrotic Remodeling of Pulmonary Vascular Bed and Improves Outcomes in the Rat Model of Chronic Thromboembolic Pulmonary Hypertension

**DOI:** 10.3390/ijms232415646

**Published:** 2022-12-09

**Authors:** Andrei A. Karpov, Aleksandra M. Mihailova, Leonid A. Shilenko, Dariya D. Vaulina, Elizaveta E. Sidorova, Anna A. Akhmetova, Pavel M. Docshin, Alexander S. Krasichkov, Kseniia E. Sanarova, Olga M. Moiseeva, Michael M. Galagudza

**Affiliations:** 1Institute of Experimental Medicine, Almazov National Medical Research Centre, 197341 Saint Petersburg, Russia; 2Center of Experimental Pharmacology, Saint Petersburg State Chemical Pharmaceutical University, 197376 Saint Petersburg, Russia; 3Department of Pathophysiology with Clinical Pathophysiology Course, First Pavlov State Medical University of Saint Petersburg, 197022 Saint Petersburg, Russia; 4Departments of Radio Engineering Systems, Saint Petersburg Electrotechnical University ‘LETI’, 197022 Saint Petersburg, Russia; 5Laboratory of Radio- and Optoelectronic Instruments for Bioinformation Technologies for Early Diagnostics of Live System Pathologies, Institute for Analytical Instrumentation, Russian Academy of Sciences, 31-33A Ivana Chernykh Street, 198095 Saint Petersburg, Russia

**Keywords:** JAK-STAT pathway, ruxolitinib, chronic thromboembolic pulmonary hypertension (CTEPH), rodent model, microspheres

## Abstract

Chronic thromboembolic pulmonary hypertension (CTEPH) is a rare complication of acute pulmonary embolism with poor clinical outcomes. Therapeutic approaches to prevention of fibrotic remodeling of the pulmonary vascular bed in CTEPH are limited. In this work, we tested the hypothesis that Janus kinase 1/2 (JAK1/2) inhibition with ruxolitinib might prevent and attenuate CTEPH in a rat model. CTEPH was induced by repeated embolization of the pulmonary artery with partially biodegradable 180 ± 30 μm alginate microspheres. Two weeks after the last injection of microspheres, ruxolitinib was administered orally at doses of 0.86, 2.58, and 4.28 mg/kg per day for 4 weeks. Prednisolone (1.475 mg/kg, i.m.) was used as a reference drug. Ruxolitinib in all doses as well as prednisolone reduced pulmonary vascular wall hypertrophy. Ruxolitinib at a dose of 2.58 mg/kg and prednisolone reduced vascular wall fibrosis. Prednisolone treatment resulted in decreased right ventricular systolic pressure. Pulmonary vascular resistance was lower in the prednisolone and ruxolitinib (4.28 mg/kg) groups in comparison with the placebo group. The plasma level of brain natriuretic peptide was lower in groups receiving ruxolitinib at doses of 2.58 and 4.28 mg/kg versus placebo. This study demonstrated that JAK1/2 inhibitor ruxolitinib dose-dependently reduced pulmonary vascular remodeling, thereby preventing CTEPH formation in rats.

## 1. Introduction

Pulmonary embolism (PE) ranks third in prevalence among all cardiovascular diseases [1,2]. It is worth considering that the risk of venous thromboembolism in patients under 45 years varies in the range of 1.0–1.5 per 1000 people per year and almost doubles every decade. This means that treatment for this population places a heavy burden on the health care system [3]. Despite adequate anticoagulant therapy, there is incomplete resolution of blood clots in 25–50% of patients with acute PE, called post-pulmonary embolism syndrome (PPES) [4]. The most severe type of PPES is chronic thromboembolic pulmonary hypertension (CTEPH). In 4–9% of patients with acute PE, the course of the disease is complicated by the development of chronic thromboembolic pulmonary hypertension (CTEPH), associated with extremely high mortality [5]. The five-year survival rate of patients with CTEPH who receive only oral anticoagulant therapy with an average pressure in the pulmonary artery (PA) (≥50 mm Hg) does not exceed 10% [6].

Currently, the CTEPH treatment algorithm is included in a multimodal approach using pulmonary thromboendarterectomy, balloon pulmonary angioplasty, and specific therapy by pulmonary circulation (PC) modern vasodilators [7]. The choice of treatment method is determined by the anatomy of the lesions [8]. Active intervention to remodel the vascular bed in the early stages after PE can reduce the risk of developing CTEPH. At the same time, pharmacological approaches based on preventing remodeling of small arteries of the pulmonary circulation (PC) in the area of embolic obstruction and beyond that results from aseptic inflammation processes and fibrosis have not been developed. An active influence on the process of vascular bed remodeling in the early stages after PE to reduce the risk of developing CTEPH could be a promising approach. The JAK-STAT pathway is one of the signaling pathways potentially involved in these processes.

Previous studies have shown that the JAK-STAT signaling pathway is an important cascade mediating the effects of a wide range of cytokines such as transforming growth factor beta (TGF-β) [9,10], platelet-derived growth factor (PDGF) [11], interleukin (IL) 6 [12], and granulocyte colony-stimulating growth factor (G-CSF) [13]. Blocking the effects of these cytokines leads to a decrease in collagen synthesis, suppression of the proliferation of smooth muscle cells, fibroblasts, endotheliocytes, and an anti-inflammatory effect. At the same time, the pathogenesis of inflammatory and autoimmune diseases is associated with the activation of JAK-STAT signaling pathway transcription. Thus, the effectiveness of JAK inhibitor (iJAK) usage during the treatment of such immune-mediated diseases as rheumatoid arthritis, psoriasis, ulcerative colitis, and graft-versus-host disease has now been confirmed [14,15]. There are no analogues of this drug for the prevention and treatment of CTEPH as yet. However, the involvement of the JAK/STAT signaling pathway in the regulation of inflammation and fibrosis of the vascular wall and lung tissue suggests its possible effectiveness in CTEPH treatment as well as other forms of lung hypertension [16,17]. The aim of this work is to study the antifibrotic effect of JAK 1,2 inhibition experimentally for the prevention and treatment of chronic thromboembolic pulmonary hypertension in rats. Prednisolone was used as a reference anti-inflammatory drug. The results of our evaluation of the morphological and functional outcomes confirmed our hypothesis about the possibility of preventing the CTEPH development by JAK/STAT pathway inhibitor.

## 2. Results

### 2.1. Animal Survival

During microsphere (MSs) administration, the mortality rate in the CTEPH group was 47% (compared with initial number of animals in CTEPH group). The causes of animals’ death in all cases were acute right ventricular heart failure and paradoxical embolism with the development of acute cerebral circulation failure. During iJAK administration, there was no loss of animals (Figure 1).

### 2.2. Treadmill Test

According to the treadmill test, 2 weeks after the last administration of MS, animals that underwent embolization of the PC vascular bed showed a significant (*p* < 0.05) decrease in exercise tolerance compared with healthy animals. All animals were divided equally into five groups according to the treadmill test results (Figure 2A). Two weeks after the start of the administration of the test substances, no significant differences between the experimental groups were found (Figure 2B). Four weeks after the start of test substance administration, the middle-dose iJAK group (IJAK 2.58) showed greater exercise tolerance compared to the high-dose iJAK group (IJAK 4.28) group (Figure 2C). There was no significant difference with the placebo (PLC) group.

### 2.3. Transthoracic Echocardiography

According to transthoracic echocardiography (TTE), there was a significant decrease in IJAK 4.28 fractional shortening (FS) compared with the intact group. No other significant differences were found between the groups (Table 1).

### 2.4. Cardiac Catheterization with Manometry

According to the right ventricle (RV) catheterization data, 4 weeks after the start of the test substances administration in the PLC group there was a significant (*p* < 0.05) increase in right ventricle systolic pressure (RVSP) compared with intact animals (Figure 3A). Furthermore, there was a significant decrease of RVSP in the prednisolone treatment group compared to the PLC group (Figure 3A). There were no significant differences between cardiac output (CO) and average LV blood pressure levels in the studied groups (Figure 3B,D). During evaluation of the RVSPCO ratio, a significant increase was determined in the PLC group compared to intact animals. However, in the PSL and IJAK 4.28 groups this ratio was significantly lower than in the PLC group (Figure 3B).

### 2.5. Histological Examination of the Lung Vessels

1883 vessels of PA branches were processed during the examination (Figure 4A). The hypertrophy index of the vascular wall of all vessels, regardless of their size, showed a significant difference between the PLC group and intact animals (*p* < 0.01). Moreover, the PSL group and low-dose iJAK group (IJAK 0.86) had significantly lower hypertrophy index values than PLC group (*p* < 0.01) (Figure 4F).

During hypertrophy indexing to evaluate various subgroups based on the outer diameter, the following data were obtained:-Vessels with mean outer diameter <100 µm: no significant differences were found between all studied groups (Figure 4B);-100–199 μm: the hypertrophy index in the PLC group was significantly higher compared to both intact animals (*p* < 0.01) and all groups treated with the tested substances (*p* < 0.01) (Figure 4C);-200–299 µm: the hypertrophy index in the IJAK 0.86 group was significantly lower compared to the PLC group (*p* < 0.01) (Figure 4D);-≥300 μm: in the IJAK 0.86 and PSL groups, the hypertrophy index was significantly lower than in the PLC group (*p* < 0.01) (Figure 4E).

To assess the severity of the vascular wall fibrosis of PA branches, 330 randomly selected vessels (55 vessels from each group) were analyzed (Figure 5A–D). In all experimental groups, the severity of fibrosis was higher compared to intact animals (*p* < 0.01). At the same time, in the PSL and IJAK 2.58 groups, the severity of fibrosis was significantly lower compared to the PLC group (*p* < 0.01) (Figure 5E).

### 2.6. Histological Heart Examination

According to the histological heart examination, a significant difference in the ratio of the areas of the RV cavity to the left ventricular (LV) cavity was determined in the PSL group compared with intact animals and the IJAK 4.28 group (*p* < 0.05, Figure 6).

### 2.7. Enzyme Immunoassay

According to enzyme immunoassay (ELISA) data, the level of IL-10 in lung tissue in the IJAK 0.86 group was significantly lower than in PSL group (*p* < 0.05). The vascular endothelial growth factor-A (VEGF-A) level in the PLC group was significantly lower than in intact animals (*p* < 0.05) (Figure 7A).

Analysis of the biologically active molecules rate in the blood plasma showed a significantly lower level of endothelin-1 in the IJAK 2.58 and IJAK 4.28 groups compared with the IJAK 0.86 group. Similar results were obtained when assessing the brain natriuretic peptide (BNP) level, with its rate significantly lower in the IJAK 2.58 and IJAK 4.28 groups compared with the PLC and PSL groups.

The IL-10 rate was significantly higher in the PLC group compared with both intact animals and the PSL IJAK 2.58 and IJAK 4.28 groups (*p* < 0.05) (Figure 7B).

## 3. Discussion

The main finding of this study is the ability of ruxolitinib and low doses of prednisolone to significantly reduce the severity of PA branch remodeling during CTEPH formation.

A CTEPH model based on partially biodegradable sodium alginate MS administration was used in this study, as this allows the most adequate pathogenesis modeling in rats [18]. Hence, all modeling pathology criteria were achieved in the PLC group [19], namely, a stable increase in systolic pressure in PC, residual obstruction of the vascular bed, decreased exercise tolerance, and conspicuous remodeling of PA branches, indicating the effectiveness of the chosen modeling protocol. During molecular analysis, a decrease in the VEGF-A rate was shown. It is possible that this is associated with endothelial dysfunction induced by aseptic inflammation and lung hypertension, providing additional confirmation of the hypothesis concerning the role of disturbed angiogenesis in CTEPH pathogenesis was [20].

The use of iJAK in this study had a positive effect on both remodeling and PC hemodynamics.

In the low-dose iJAK group (IJAK 0.86), there was a decrease in the vascular wall hypertrophy index among all PA branches, regardless of their size, as well as in subgroups of vessels with an average outer diameter of 100–199 μm, 200–299 μm, and ≥ 300μm. The decrease of IL-10 in lung tissue compared with the placebo group is possibly associated with the immunosuppressive properties of ruxolitinib mediated through other molecular pathways, such as suppression of the synthesis of IL-2, IL-4, IL-6, etc. [21].

The middle-dose iJAK group (IJAK 2.58) was characterized by decreasing of the vascular wall hypertrophy index in subgroups of vessels with an average outer diameter-100–199 μm and ≥300 μm as well as by a significant decrease in collagen fiber percentage in the vessel wall compared with the PLC group. The volume of collagen fiber deposition on the vascular wall was directly related to irreversible vessel fibrous remodeling. This is one of the key indicators that determine the prognosis of the disease course. The average iJAK dose influence on aseptic inflammation and vascular wall remodeling was represented in the cytokine profile, where the endothelin-1 rate decreased. This is one of the most adequate markers of endothelial dysfunction [22], along with BNP, which directly correlates with the severity of cardiac chamber overload [23]. A decrease in the IL-10 rate in blood plasma, as in the IJAK 0.86 group, can be ascribed to other immunosuppression mechanisms.

In the high-dose iJAK group (IJAK 4.28), a significant decrease in the vascular wall hypertrophy index was shown in the vessels’ subgroup, with an average outer diameter 100–199 µm. During hemodynamic evaluation, a decrease in the RVSP/CO ratio was revealed, which may be associated with decreasing pulmonary vascular resistance. Remarkably, the remodeling of small-diameter vessels had the most significant effect on vascular resistance formation. However, despite the positive effects of the high iJAK dose, this group had significantly lower exercise tolerance compared to the IJAK 2.58 group. It should be pointed out that exercise tolerance is an integral marker of the body. It is influenced by a number of parameters, such as PC hemodynamics, systemic hemodynamics, and the presence of general intoxication.

A significant decrease in exercise tolerance in the high-dose iJAK group may be directly related to LV systolic function decreasing, identified by TTE. While the mechanism of this decrease is unknown, a direct cardiotoxic effect of a high drug dose cannot be excluded. When analyzing sources in the literature, a single description of ruxolitinib cardiotoxicity was found. This was registered in clinical practice and accompanied by a decrease in LV systolic function [24].

However, in contrast to these data, in the IJAK 4.28 group the BNP level in plasma was significantly lower than in the PLC group, and CO remained within the normal range, which queried significant toxic myocardial damage. Thus, the revealed change in the shortening fraction requires further study.

Taking into account that nowadays there are no drugs for CTEPH treatment that are used directly for fibrosis suppression [25,26], prednisolone at a low dose (1.475 mg/kg, corresponding to 0.25 mg/kg in humans) was chosen as a reference drug to investigate the potential for suppressing aseptic inflammation. There are only a few studies that have focused on testing glucocorticosteroids in CTEPH treatment. Kerr K.M. et al. [27] applied a short course of methylprednisolone used immediately prior to pulmonary thromboendarterectomy to prevent acute lung injury. A decrease in IL-6 and IL-8 levels was demonstrated along with an increase in IL-10 in blood plasma, as well as a decrease in IL-1 and IL-6 in bronchoalveolar lavage (BAL fluid) one day after surgery. However, these molecular changes did not impact the frequency of acute lung injury. Glucocorticosteroids suppress inflammation by preventing the activation of the transcription factors nuclear factor-kappa B and activator protein-1, mainly by reversing histone acetylation of activated inflammatory genes through binding of liganded glucocorticoid receptors to coactivators and recruitment of histone deacetylase-2 (HDAC2) to the activated transcription complex. At higher concentrations of corticosteroids, glucocorticoid receptor homodimers interact with DNA recognition sites, activating transcription of anti-inflammatory genes [28]. In this study, the use of prednisolone led to a significant improvement in hemodynamic parameters, specifically, a decrease in RVSP and RVSP/CO ratio. Moreover, there was a decrease in the vascular wall hypertrophy index, both among the vessels’ total volume and in subgroups of vessels with an average outer diameter of 100–199 μm, ≥300 μm. The severity of collagen fiber deposition was significantly lower than in the PLC group. However, during the cytokine profile analysis, the BNP level was significantly higher than in the IJAK treatment groups and did not differ from the PLC group. The absence of any decrease in endothelin-1 level in plasma may indicate an insufficient effect of prednisolone on endothelial dysfunction. Interesting data were obtained when assessing the severity of cardiac remodeling; the ratio of the RV to the LV cavity area was the highest among all studied groups, and significantly higher than in intact animals and the IJAK 4.28 group. A subsequent detailed study is required to determine the reasons for the significant RV dilatation after prednisolone administration.

Thus, iJAK usage in this study had a positive effect on both PC remodeling and PC hemodynamics, as well as on the cytokine profile of molecules responsible for inflammation and dysfunction in the cardiovascular system. Previously, iJAKs have not been used to prevent and treat CTEPH, which makes it difficult to compare obtained data with other sources. However, in a recent article from Yerabolu D. et al. [16], the role of the JAK-STAT pathway of two different forms of pulmonary hypertension formation has been experimentally demonstrated through the induction of hypoxia in mice and monocrotaline in rats. In this study, ruxolitinib reduced pulmonary hypertension and RV remodeling in both experimental models. These data may circumstantially indicate the prospects for iJAK use in various forms of pulmonary arterial hypertension. Interesting data from a comprehensive bioinformatics analysis of three large databases of human lung tissues have been obtained as well [29]. Mathematical modeling and experimental work on a smooth muscle cell culture (SMC) of PA has shown the ability of ruxolitinib to suppress the proliferation and migration of vascular wall SMC, thereby affecting the remodeling of the vascular wall in pulmonary hypertension. Moreover, this study experimentally confirmed numerous works carried out in vitro, revealing the spectrum of iJAK action mechanisms on fibrosis formation and aseptic inflammation with different localizations.

The role of the JAK-STAT pathway in the regulation of IL-2, 4, 6, 12, 13, 17, 21, 23, and 31 [21,30] as well as factor growth fibroblast-2 (FGF-2) [21], TGF-β [9,10], and PDGF has been shown [11]. Thus, based on the obtained data and on the already-known mechanisms of JAK 1,2 inhibitor action, the key effect of ruxolitinib in this study was its effect on the remodeling of PA branches.

## 4. Materials and Methods

All studies were carried out in accordance with the Care and Use of Laboratory Animals Guidelines (National Institutes of Health publication, 8th ed., 2011). Procedures with animals were reviewed and approved by the bioethical commission of the Federal State Budgetary Educational Institution of Higher Education “SPCPhU” of the Ministry of Health of the Russian Federation.

### 4.1. Animals

Male Wistar rats of conventional rank with an average weight of 225 ± 28 g were used in this study. The animals were maintained in standard barrier vivarium conditions and were provided with food and water ad libitum.

### 4.2. Embolic Particles

Partially biodegradable sodium alginate MSs were used as embolic particles in this study. MSs were obtained from ultrapure sodium alginate (SigmaAldrich, St. Louis, MO, USA) using a B-390 electrostatic encapsulator (Buchi, Switzerland) [18]. A 2% barium chloride solution was used as a stabilizing agent. The size of the resulting size MSs was 180 ± 30 µm. All microspheres were produced under sterile conditions.

### 4.3. JAK Inhibitor Dose Calculation

Ruxolitinib (MedChemExpress LLC, Princeton, NJ, USA) was used as a JAK 1,2 inhibitor. The studied dose calculations for laboratory animals were based on the range of drug doses for human usage. For various indications, this was from 10 to 50 mg per day. For conversion, the following ratio was used: (drug dose/average human weight (70 kg)) × 5.9 according to which we determined a following range of 0.86–4.28 mg/kg per day [31]. Similar doses of ruxolitinib have been used in other experimental studies in rats to demonstrate the renoprotective effect of ruxolitinib in a model of diabetic nephropathy [32]. In light of the potential dose-dependence of the ruxolitinib effect, we evaluated its effects at three different daily doses within the above range, specifically, 0.86, 2.58, and 4.28 mg/kg. The duration of ruxolitinib administration was 4 weeks, starting from the second week after the last injection of MSs.

### 4.4. Experimental Protocol

MSs (V = 50 µL) were suspended in 1 mL of saline and were injected eight times at 4-day intervals. Two weeks later, the day after the last injection of MSs, a treadmill test was performed; the animals were equally divided into five groups such that there were no significant differences in exercise tolerance between the formed groups. The intact animal group was used as a negative control (Figure 8).

### 4.5. Animal Groups

Intact animals (INT), *n* = 11Placebo (PLC), *n* = 11: two weeks after the last injection of MSs, animals were injected intravenously with saline daily over 4 weeks.Prednisolone (PSL), *n* = 10: two weeks after the last administration of MSs, prednisolone at a dose of 1.475 mg/kg (which is equivalent to 0.25 mg/kg in humans) was administered intramuscularly over 4 weeks.Low-dose iJAK (IJAK 0.86), *n* = 11: two weeks after the last administration of MSs, iJAK was administered at a daily dose of 0.86 mg/kg (divided into two intakes) over 4 weeks per os (equivalent to 10 mg per day in humans).Middle-dose iJAK (IJAK 2.58), *n* = 11: two weeks after the last administration of MSs, iJAK was administered at a daily dose of 2.58 mg/kg (divided into two intakes) over 4 weeks per os (which is equivalent to 30 mg per day in humans).High-dose iJAK (IJAK 4.28), *n* = 10: two weeks after the last administration of MSs, iJAK was administered at a daily dose of 4.28 mg/kg (divided into two intakes) during 4 weeks per os (which is equivalent to 50 mg per day in humans).

Two and four weeks after administration of the studied substances, a treadmill test was performed to assess changes in exercise tolerance during the treatment.

Four weeks after iJAK therapy, the morphological and functional parameters of the cardiovascular system were studied, specifically, echocardiography, RV catheterization with manometry, and histological studies of the heart and lungs.

### 4.6. Research Design

Treadmill Test

Treadmill tests were performed in the 2 weeks after the last administration of MSs, before dividing animals into groups, and in the 2 and 4 weeks after administration of the studied substances. A treadmill device (model LE8710, Harvard Apparatus, Holliston, MA, USA) was used to perform the stress test. During testing, a protocol with a gradual increase in the rotation speed of the treadmill belt was used: 5 m/min, increasing every 30 s up to a speed of 40 m/min. The distance that each animal ran during testing was assessed (m). Exercise tolerance testing along with echocardiography and manometry were conducted by investigators blinded to animal groups.

2.Transthoracic Echocardiography

TTE was performed 4 weeks after administration of iJAK. To perform the study, a high-resolution ultrasound unit (MyLab One Touch SL 3116, Esaote, Genoa, Italy) with a vascular linear probe (frequency: 13 MHz, scanning depth: 2 cm) was used. During TTE, the animals were anesthetized via isoflurane inhalation using a SomnoSuite Low-Flow Anesthesia System for gas anesthesia (Kent Scientific, Torrington, CT, USA) and placed on a heated table (TCAT-2LV controller, Physitemp Instruments Inc., Clifton, NJ, USA) in the supine position. The main parameters evaluated in the study were: (1) the diameter of the main PA (mm); (2) peak flow rate in the MPA outflow tract (Vmax MPA, m/s); (3) tricuspid annular plane systolic excursion (TAPSE, mm); and (4) LV FS (%).

3.Cardiac Catheterization with Manometry

Cardiac catheterization with RV pressure measurement took place 4 weeks after administration of iJAK. Before cardiac catheterization, the animals were prepared for studies as well as for the TTE procedure. Artificial lung ventilation was carried out via pulmotor SAR—830/AP (CWE Inc., USA). The following parameters of artificial lung ventilation were used: respiratory rate 60/min and respiratory capacity 3 mL/100 g of body weight. RV manometry was performed through RV apex puncture. The registration of hemodynamic parameters (mean blood pressure (BP), systolic, diastolic, and mean RV pressure) was performed using the PhysExp Mini recorder (Cardioprotect Ltd., St. Petersburg, Russia). To register CO, a TS420 Perivascular Flow Module volume flow sensor (Transonic, Ithaca, NY, USA) was inserted into the ascending part of the aorta. The ratio of RVSP/CO was used to evaluate pulmonary vascular resistance [33,34].

4.Histological Examination

Histological studies were performed 4 weeks after administration of the tested substances. The animals were euthanized with potassium chloride given intravenously under deep isoflurane anesthesia (10% solution, 1 mL, i.v.). The right lower lobe was used for histological evaluation. It was divided into two lateral levels for analysis. Slices 3–5 µm thick were stained with hematoxylin and eosin and with Picro-Mallori stain (Biovitrum, St. Petersburg, Russia) for connective tissue. The tissue specimen evaluation was using an Eclipse Ni-U (Nikon, Japan) with magnification from ×5 to ×40. Microscopy results were evaluated using Nis Elements Br4 software (Nikon, Japan). The average outer diameter of the vessel and the hypertrophy index, which is the ratio of the vascular wall area to the total vessel area in percent, were determined on two distal lung sections in all identified vessels belonging to the PA branches [18]. For data interpretation, all analyzed vessels were divided into groups according to outer diameter: <100 µm, 100–199 µm, 200–299 µm, and ≥300 µm; hypertrophy index was determined separately for each subgroup. To assess the severity of fibrous remodeling of the PA branches, a special index was calculated based on the percentage of collagen fibers of the vascular wall using the Python plugin for fibrosis and IGH assessment 1.1 (ETU “LETI”, Saint Petersburg, Russia).

To study the RV structure, the heart below the left ventricular appendage was cut into three equal transverse portions; sections 4–5 µm thick were stained with hematoxylin and eosin. The ratio of the area of the RV cavity to the area of the LV cavity was used as the criterion for RV remodeling.

### 4.7. Enzyme Immunoassay

Sampling of lung tissue and blood plasma to evaluate the level of biologically active molecules was performed immediately after the animals’ euthanasia using an enzyme immunoassay. The following parameters were determined in the lung tissue: IL-1, IL-6, IL-10, and VEGF-A; in blood plasma, IL-1, IL-10, endothelin-1, and BNP were determined. The study was performed using an automated biochemistry analyzer and an ELISA analyzer (ChemWell Combi 2910, Awareness Technology, Palm City, FL, USA) using commercial kits.

### 4.8. Data Analysis

All data are expressed as mean ± standard deviation. In terms of the small group size and the absence of a normal value distribution, non-parametric statistical methods were used for statistical data processing. Specifically, the Kruskal-Wallis test was used to evaluate the significance of differences between two mismatched populations.

Data analysis was carried out using the Statistica v10.0 (StatSoft, Tulsa, OK, USA) package, with *p* values ≤ 0.05 considered to indicate statistical significance.

## 5. Conclusions

Our results demonstrate a dose-dependent ability of ruxolitinib to reduce the severity of PC remodeling and increase exercise tolerance during CTEPH formation in rats was. These data indicate a significant role of aseptic inflammation and following fibrosis in the formation and progression of CTEPH. Furthermore, iJAK (as a drug class) can be used in the complex treatment and prevention of CTEPH.

## Figures and Tables

**Figure 1 ijms-23-15646-f001:**
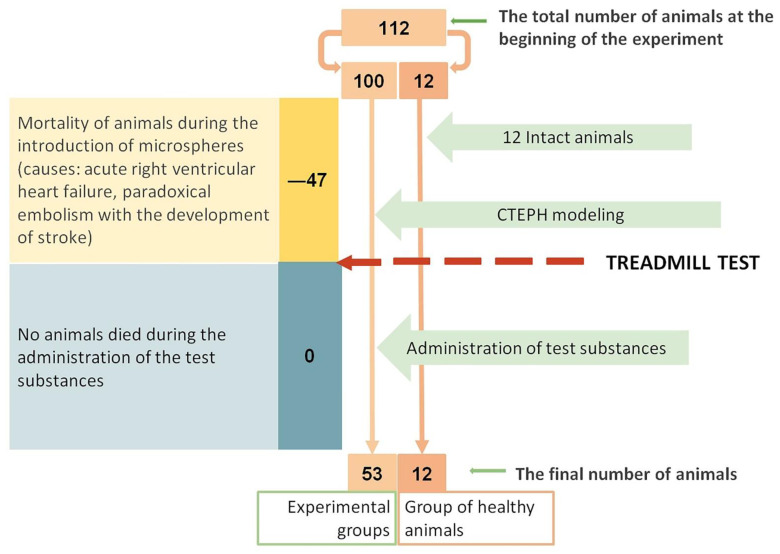
Scheme of loss of experimental animals.

**Figure 2 ijms-23-15646-f002:**
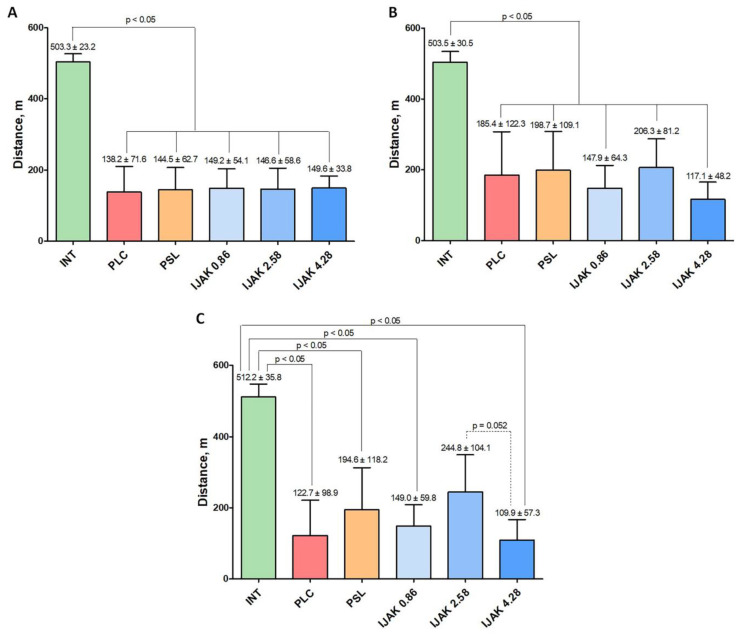
The results of non-invasive tests using the treadmill test. (**A**) Evaluation of exercise tolerance two weeks after the last injection of microspheres, (**B**) two weeks after the start of the test substances administration, and (**C**) four weeks after the start of test substance administration. INT—intact animals; PLC—placebo; PSL—prednisolone; IJAK—JAK inhibitor.

**Figure 3 ijms-23-15646-f003:**
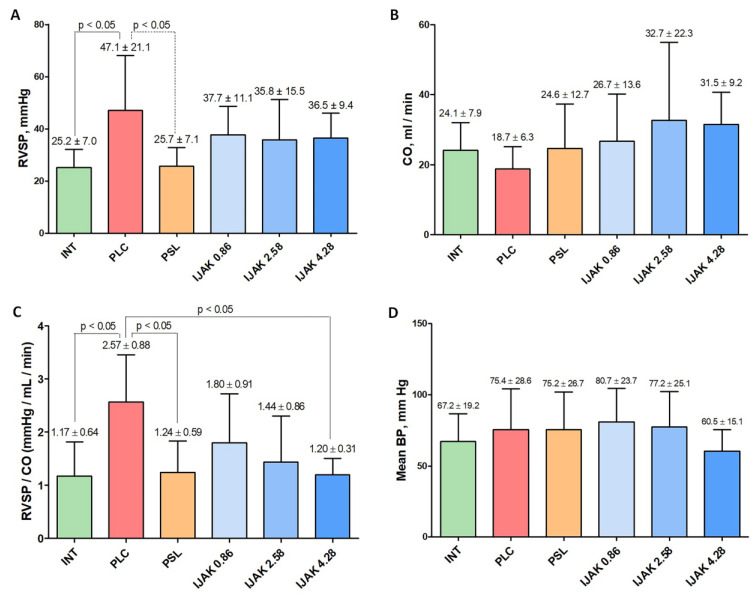
Hemodynamic parameters 4 weeks after the start of the test substances administration according to the data of cardiac catheterization with manometry. (**A**) Right ventricular systolic pressure (RVSP); (**B**) Cardiac output (CO); (**C**) RVSP/CO ratio; (**D**) Mean blood pressure (BP). INT—intact animals; PLC—placebo; PSL—prednisolone; IJAK—JAK inhibitor.

**Figure 4 ijms-23-15646-f004:**
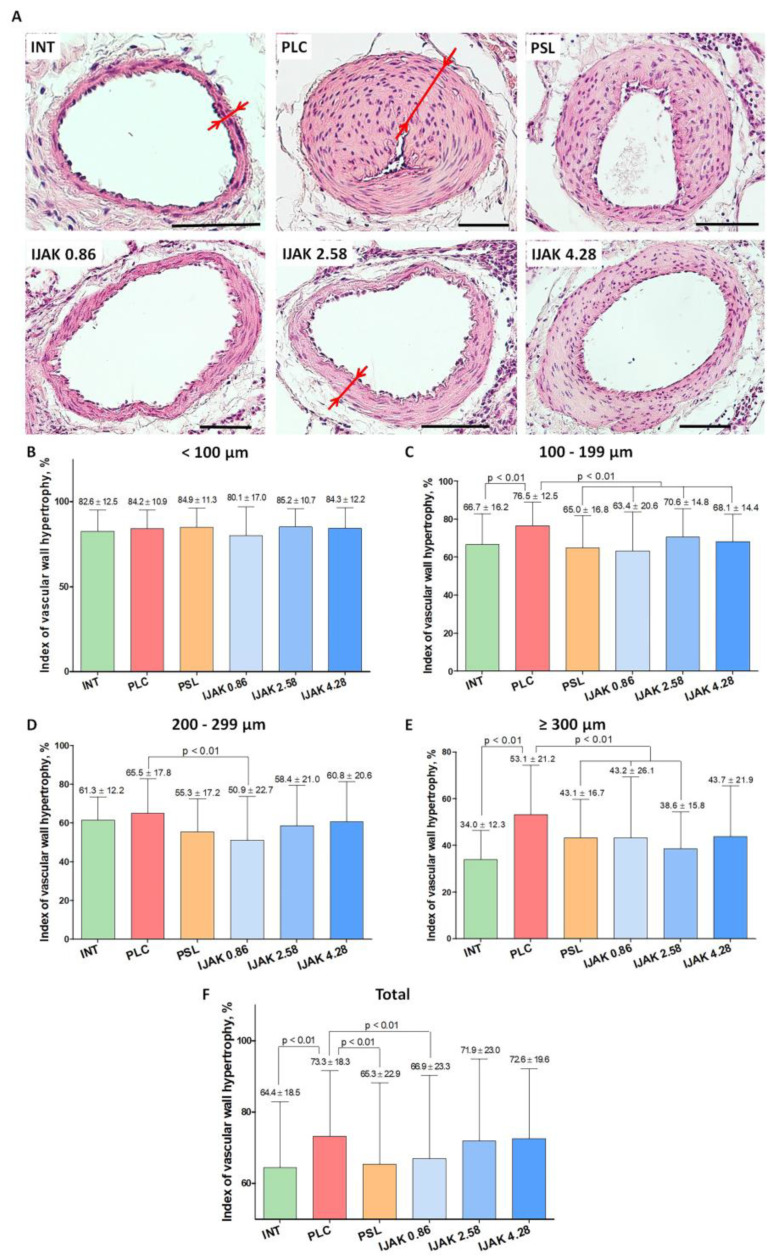
Histological examination of lung vessels. (**A**) Representative micrographs of PA branches in studied groups (staining: hematoxylin–eosin, scale bar: 100 µm; red arrows indicate the thickness of the vascular wall). (**B**–**F**) Hypertrophy index of the vascular wall of PA branches: (**B**) vessels with outer diameter <100 µm; (**C**) vessels with outer diameter 100–199 µm; (**D**) vessels with outer diameter 200–299 µm; (**E**) vessels with outer diameter ≥ 300 µm; (**F**) all vessel diameters. INT—intact animals; PLC—placebo; PSL—prednisolone; IJAK—JAK inhibitor.

**Figure 5 ijms-23-15646-f005:**
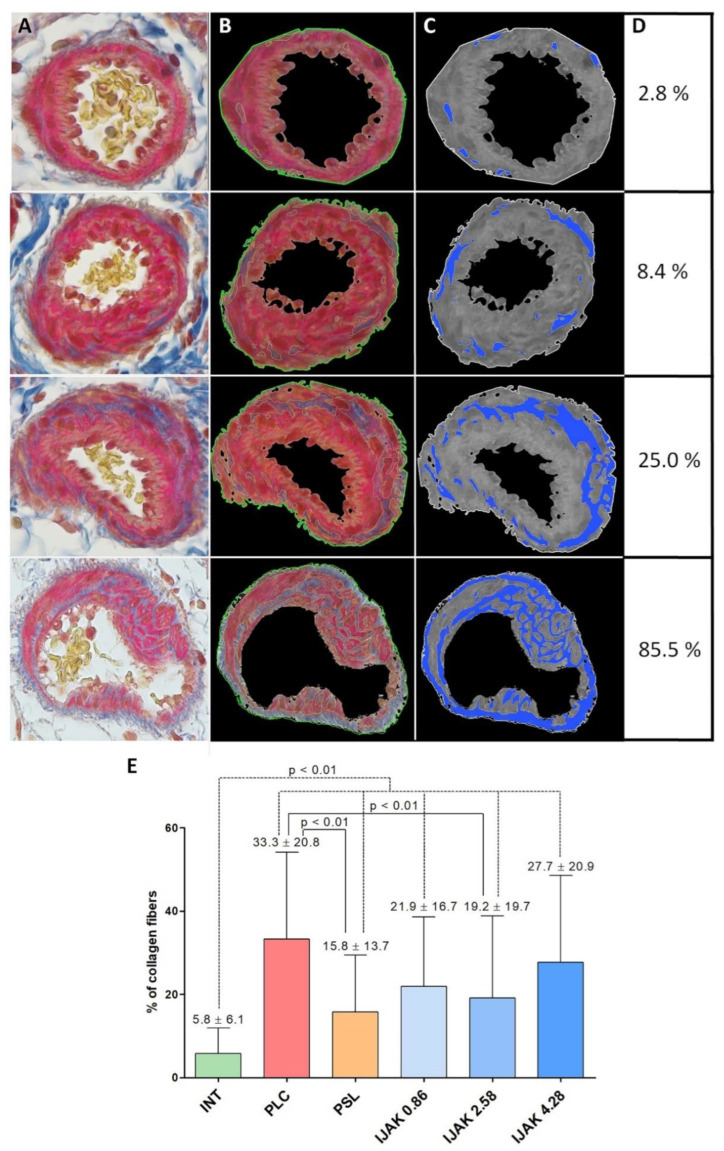
The evaluation of the vascular wall fibrosis of PA branches. (**A**–**C**) Representative micrographs of vessels: (**A**) Picro-Mallory staining; (**B**) vessel border selection by Python plugin for fibrosis and IGH assessment 1.1 (ETU “LETI”, Saint Petersburg, Russia); (**C**) monochrome isolation of collagen fibers in the vascular wall structure; (**D**) nominal percentage of the vascular wall fibrosis of representative micrographs; (**E**) percentage of collagen fibers in the vascular wall structure of the PA branches in the studied groups. INT—intact animals; PLC—placebo; PSL—prednisolone; IJAK—JAK inhibitor.

**Figure 6 ijms-23-15646-f006:**
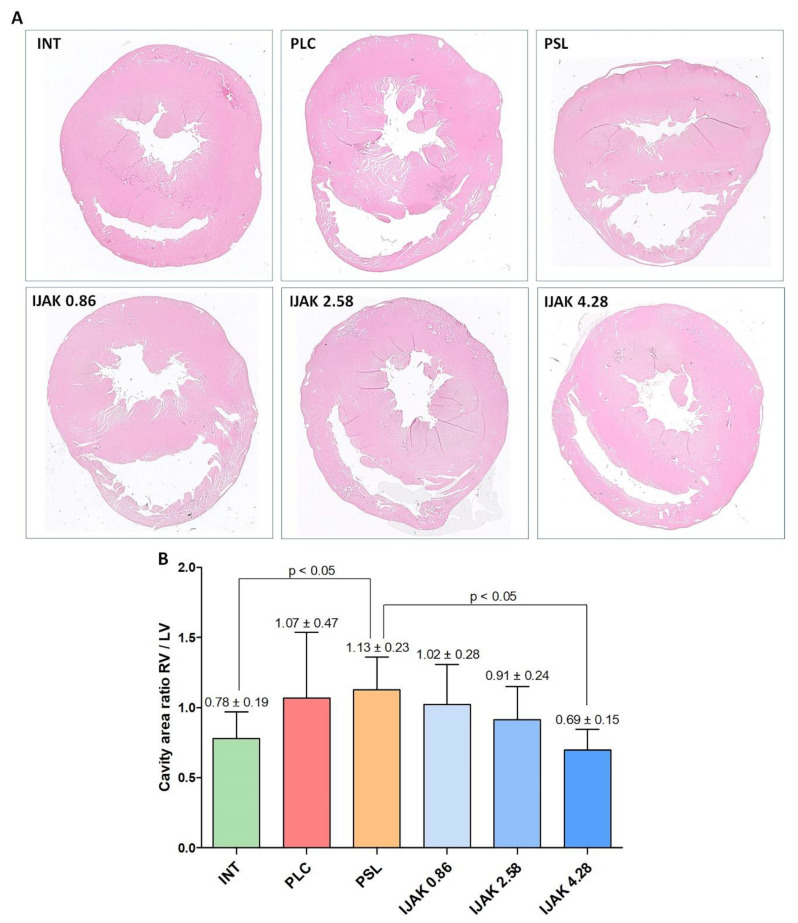
Histological heart examination. (**A**) Representative transverse heart section of study groups stained by hematoxylin–eosin; (**B**) ratio between RV cavity area and LV cavity area. INT—intact animals; PLC—placebo; PSL—prednisolone; IJAK—JAK inhibitor.

**Figure 7 ijms-23-15646-f007:**
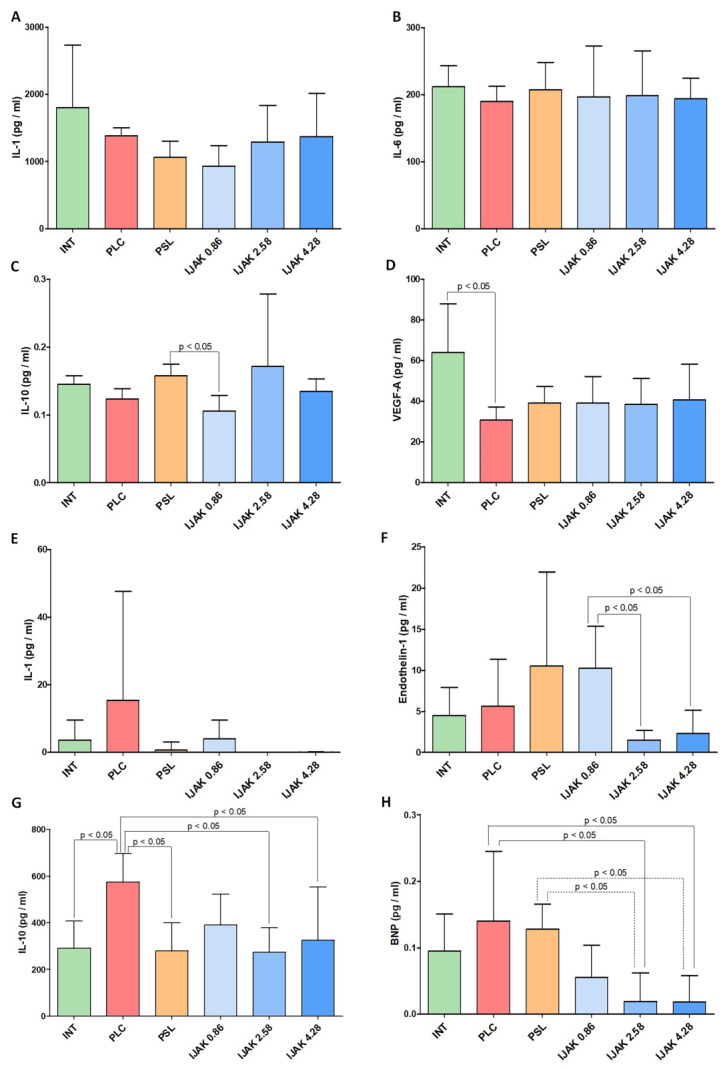
Results of enzyme immunoassay of bioactive molecules rate (**A**–**D**) in lung tissue and (**E**–**H**) in blood plasma. INT—intact animals; PLC—placebo; PSL—prednisolone; IJAK—JAK inhibitor.

**Figure 8 ijms-23-15646-f008:**
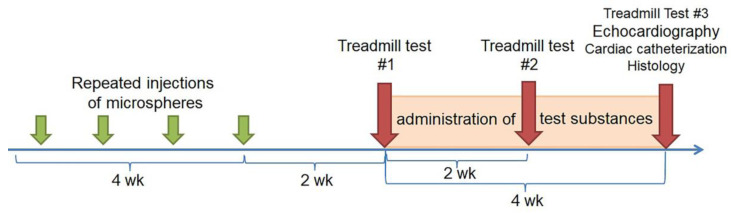
Scheme of the study design.

**Table 1 ijms-23-15646-t001:** Results of echocardiographic examination 4 weeks after the start of administration of substances during this study.

Parameters	INT	PLC	PSL	IJAK 0.86	IJAK 2.58	IJAK 4.28
PT diameter (mm)	2.87 ± 0.23	2.98 ± 0.28	3.07 ± 0.32	2.88 ± 0.26	3.14 ± 0.4	2.96 ± 0.24
RVOT diameter (mm)	3.79 ± 0.15	3.95 ± 0.33	3.73 ± 0.26	3.85 ± 0.19	3.94 ± 0.19	4.0 ± 0.4
Vmax in PT (sm/s)	346 ± 58	313 ± 41	314 ± 19	331 ± 39	340 ± 26	335 ± 40
Vmax RVOT (sm/s)	255 ± 49	224 ± 35	221 ± 36	232 ± 21	235 ± 30	231 ± 29
HR (beats/min)	286 ± 28	248 ± 66	245 ± 25	272 ± 52	259 ± 24	251 ± 62
LV EF (%)	59 ± 11	58 ± 11	57 ± 7	50 ± 7	52 ± 9	47 ± 7 *
TAPSE (mm)	2.34 ± 0.39	2.11 ± 0.63	1.87 ± 0.46	2.22 ± 0.58	2.06 ± 0.37	2.42 ± 0.73

*—significant (*p* < 0.05) difference compared to intact animals. LV—left ventricle, PT—pulmonary trunk, RV—right ventricle, HR—heart rate, RVOT—RV outflow tract, FS—fractional shortening, TAPSE—tricuspid annular plane systolic excursion; INT—intact animals; PLC—placebo; PSL—prednisolone; IJAK—JAK inhibitor.

## Data Availability

Not applicable.

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
