# Peer review of "Inhibition of JAK1,2 Prevents Fibrotic Remodeling of Pulmonary Vascular Bed and Improves Outcomes in the Rat Model of Chronic Thromboembolic Pulmonary Hypertension"

_ijms, 2022, doi:10.3390/ijms232415646_

Round 1
Reviewer 1 Report
The antifibrotic effect of ruxolitinib has already been demonstrated in many studies in some other diseases. This study provides new results when it comes to thromboembolic pulmonary hypertension, indicating the potential use of ruxolitinib as a drug in the case of this disease as well. In my opinion, there is no need for any further improvement of the methodology in this paper. The conclusions are in line with the initial hypothesis. The results are clearly presented in figures and pictures. The only recommendation is to double-check the spelling.
Author Response
We would like to thank the reviewer for the constructive comments on the manuscript.
Point 1: The reviewer asked to check the spelling in the text.
Response: We have checked the spelling in the text, all changes and reformulations were highlighted.
Reviewer 2 Report
The present manuscript is a well-designed, well-executed, well-written research article, the authors did a great study here! I would like to thank them for their great efforts.
However, I have the following comments in order to improve the manuscript.
Please enlarge the figures and improve the resolution and size of the text to make them clearly visible to readers of all ages.
Figure 4A, please show the differences in the figures using special characters (arrows, etc.), which color stains what etc. to make them easy to understand.
Same for Figure 5A-C, and 6A.
Please include a conclusion section based on the results obtained.
Please include a graphical abstract or a summary figure for a better illustration of the study findings.
Author Response
We would like to thank the reviewer for the constructive comments on the manuscript.
Point 1: The reviewer asked to enlarge all the figures and improve the resolution and size of the text.
Response: We have changed all figures in the text according to the reviewer’s improvements.
Point 2: The reviewer asked to show the differences in the figures (4A, 5A-C, 6A) using special characters (arrows) to make them easy to understand.
Response: We have improved named figures to make the easy to understand.
Point 3: The reviewer asked to include a conclusion based on the results obtained.
Response: We have added a conclusion section.
Point 4: The reviewer asked to include a graphical abstract or a summary figure for a better illustration of the study findings.
Response: We have added a graphical abstract.